# Helix breaking transition in the S4 of HCN channel is critical for hyperpolarization-dependent gating

Marina A Kasimova[1†], Debanjan Tewari[2†], John B Cowgill[2,3†],
Willy Carrasquel Ursuleaz[2], Jenna L Lin[2,3], Lucie Delemotte[1*], Baron Chanda[2,4*]

[1]Science for Life Laboratory, Department of Applied Physics, KTH Royal Institute of Technology, Stockholm, Sweden; [2]Department of Neuroscience, University of Wisconsin-Madison, Madison, United States; [3]Graduate program in Biophysics, University of Wisconsin, Madison, United States; [4]Department of Biomolecular Chemistry, University of Wisconsin-Madison, Madison, United States

**Abstract** In contrast to most voltage-gated ion channels, hyperpolarization- and cAMP gated (HCN) ion channels open on hyperpolarization. Structure-function studies show that the voltage-sensor of HCN channels are unique but the mechanisms that determine gating polarity remain poorly understood. All-atom molecular dynamics simulations (~20 μs) of HCN1 channel under hyperpolarization reveals an initial downward movement of the S4 voltage-sensor but following the transfer of last gating charge, the S4 breaks into two sub-helices with the lower sub-helix becoming parallel to the membrane. Functional studies on bipolar channels show that the gating polarity strongly correlates with helical turn propensity of the substituents at the breakpoint. Remarkably, in a proto-HCN background, the replacement of breakpoint serine with a bulky hydrophobic amino acid is sufficient to completely flip the gating polarity from inward to outward-rectifying. Our studies reveal an unexpected mechanism of inward rectification involving a linker sub-helix emerging from HCN S4 during hyperpolarization.

*For correspondence:
lucie.delemotte@scilifelab.se (LD);
chanda@wisc.edu (BC)

[†]These authors contributed equally to this work

## Introduction

The hyperpolarization-activated and cyclic nucleotide activated ion channels are found in the pace-making cells of heart and brain where they play a singular role in regulating rhythmic electrical oscillations (*Brown et al., 1979*; *Ludwig et al., 1998*; *Santoro et al., 1998*). Unlike other members of the voltage-gated ion channel family, these channels open at membrane potentials below the threshold of action potential and depolarize the membrane by increasing the permeability to Na$^+$ ions (*Brown et al., 1979*). Their slow gating kinetics and low permeability determine the frequency of action potential spikes (*Larson et al., 2013*; *Sharpe et al., 2017*). In addition to voltage, the gating properties of these channels are regulated by PIP$_2$ and second messengers such as cAMP which mediates the 'fight or flight' response by increasing the activity of HCN channels in sinoatrial nodal cells of the heart (*Brown et al., 1979*; *Pian et al., 2006*).

From a structural standpoint, the HCN channels are remarkably similar to other voltage-gated ion channels. They are tetrameric and each subunit consists of a voltage-sensing domain (VSD) (S1-S4 helices) and pore-forming domain (S5-S6 helices) (*Lee and MacKinnon, 2017*). Like other members of CNBD family, the VSD and pore domain (PD) are arranged in a non-domain swapped arrangement (*Lee and MacKinnon, 2017*; *Whicher and MacKinnon, 2016*). The C-linker region connects the cyclic nucleotide binding domain (CNBD) in the carboxy terminal end to the S6 transmembrane segment. The resting state structure of the human HCN1 channel (hereafter referred to as HCN1) reveals several distinctive features. First, the S4 segment of HCN1 channel is at least two helical

turns longer than the corresponding S4 in depolarization activated ion channels. Second, at the cytosolic amino terminal end, a 45 residue stretch forms a small 3-helix bundle which interacts with the voltage-sensor and CNBD domain. This domain is unique to this channel family and is called the HCN domain. Finally, the S4 helix of the HCN1 channel is in tight juxtaposition with the S5 helix, unlike in the related depolarization activated ether-á-go-go (EAG) channel (*Lee and MacKinnon, 2017*).

Recent studies show that the voltage-sensing domains of the HCN channels have an intrinsic ability to drive the channel in both hyperpolarizing and depolarizing directions (*Cowgill et al., 2019*). This ability to open the channel on hyperpolarization neither depends on the extra length of the S4 segment nor on the HCN domain suggesting that there are other non-obvious determinants of gating polarity in the voltage-sensing S4 of HCN channels. Cysteine accessibility studies to probe the nature of these conformational changes show that the S4 undergoes a large downward motion much like the voltage-sensor of the Shaker potassium channel (*Bell et al., 2004*; *Männikkö et al., 2002*; *Vemana et al., 2004*). One notable difference, however, is that the residues in the top part of the S4 helix showed larger displacements compared to those at the lower end. Recently, transition metal FRET studies also propose that the S4 undergoes a downward motion with a slight bend at the tail end of the S4 helix (*Dai et al., 2019*). Nevertheless, it remains unclear whether these subtle differences in the voltage-sensor motions between outward (depolarization-activated) and inward rectifying (hyperpolarization-activated) channels can account for the distinct gating phenotypes.

In order to probe the structural transitions through the gating cycle, we took advantage of the special purpose Anton 2 supercomputer (*Shaw et al., 2014*) to carry out two independent runs of tens of microseconds long molecular dynamics (MD) simulations. Our simulations showed that the downward movement of the voltage-sensors is accompanied by breaking of the S4 helix into two sub-helices and the lower sub-helix adopting an orientation parallel to the membrane plane and turning into a surrogate S4-S5 linker. To probe the importance of this helix breaking transition, we utilized bipolar chimeras which exhibit both hyperpolarization and depolarization-activated currents (*Cowgill et al., 2019*). Many of the mutations at this region in the wild type background result in a non-functional phenotype. The bipolar background enables us to robustly quantify the relative contributions of various molecular manipulations on hyperpolarizing vis-à-vis depolarizing currents. Our studies reveal that both the location of the breakpoint and nature of amino acid therein is critical for gating polarity in the channels of this superfamily.

## Results

### Atomistic simulations of HCN1 voltage sensor activation

HCN channels are notoriously slow-activating such that even the fastest family members require tens of milliseconds to open. Because this timescale is not accessible in even the most advanced simulations, we employed two strategies to accelerate channel kinetics. First, we applied strongly hyperpolarizing conditions (−550 mV) similar to the approach used for simulations of Kv1.2/2.1 deactivation (*Jensen et al., 2012*). Next, we sought to facilitate the movement of gating charges by further hydrating water crevices inside the voltage-sensing domain (VSD). Inspired by the work of *Lacroix et al. (2013)*, we designed a mutant that was anticipated to increase hydrophilicity in the VSD. By comparing the sequences of HCN and EAG homologs, which activate and gate much faster, we identified two positions in S1 with large hydrophobic residues in HCN channels compared to smaller, less hydrophobic residues in the EAG family (*Figure 1—figure supplement 1A,C*). Based on this information, we ran preliminary simulations (~3 µs) on the wild-type and mutant (M153T/I160V) HCN1 under hyperpolarizing electric field. In agreement with our prediction, the mutant showed signs of faster activation than the wild type (WT) channel (*Figure 1—figure supplement 1B*), therefore we proceeded to run Anton simulations with the mutant channel. Functional testing shows that these mutations are well tolerated (*Figure 1—figure supplement 1D*).

Substantial conformational change was observed in 6 out of 8 voltage sensors from two independent ~20 µs simulations of the full length channel (*Figure 1A,B*). A large downward movement of the S4 helix drives two positively charged residues across the focused electric field at the charge transfer center (F186) (*Figure 1A,C*). This large vertical movement agrees with previous predictions based on cysteine accessibility (*Bell et al., 2004*; *Vemana et al., 2004*) and voltage clamp

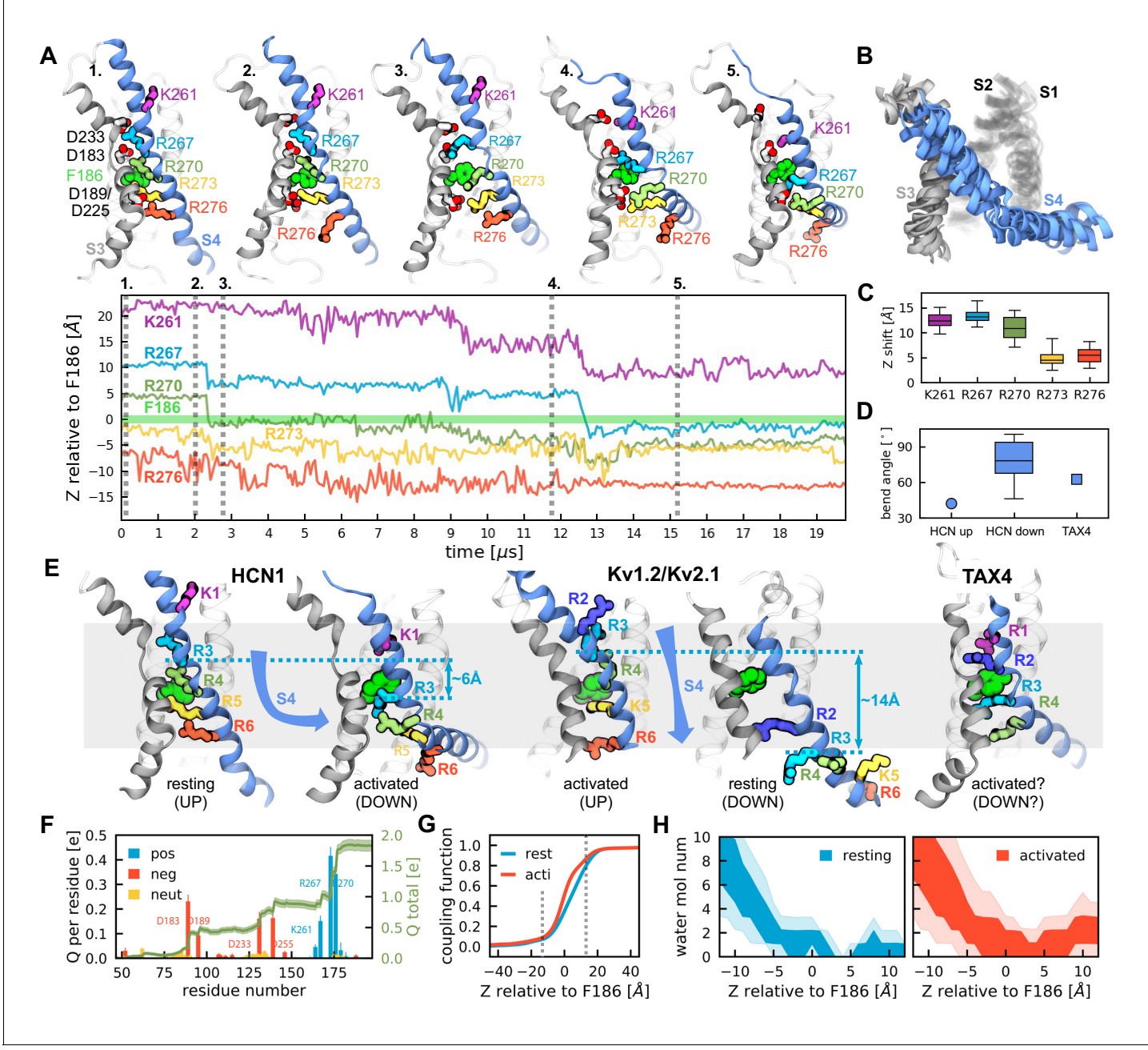

**Figure 1.** Activation of HCN1 via multi-microsecond long MD simulations. (**A**) (Top) Conformational changes of the HCN1 voltage sensor triggered by an applied electric field. Representative snapshots of one voltage sensor along the trajectory are shown. The numbers correspond to the timepoints marked on the trajectory plot below. S4 is shown in blue, S3 in gray, S1 and S2 are transparent. (Bottom) Time-dependent displacement of the gating charges along the membrane normal in a representative voltage sensor measured relative to the charge transfer center (F186). (**B**) Overlay of representative structures of six activated voltage sensors from two independent ~20 μs simulation runs. (**C**) Overall displacement of gating charges (S4 basic residues) along the electric field direction. The box plot shows the median, 25-75% (box), 1-99% (bars) of the data collected from the six voltage sensors that underwent activation. (**D**) A comparison of the bend angles of lower S4 sub-helix with the principal axis of the VSDs of HCN1 and TAX4. HCN1 Up shows the angle between the S4 C-terminus of the HCN1 (PDB 5U6O) and the principal axis of its VSD; HCN1 Down shows the angle between the S4 C-terminus of the simulated structures at the end of the run and the principal axis of its VSD; TAX4 shows the angle between the S4 C-terminus of the TAX4 channel (PDB 5H3O, *Li et al., 2017*) and the principal axis of its VSD. The box plot shows the median, 25-75% (box), 1-99% (bars) of the data collected from the six voltage sensors that underwent activation. (**E**) Comparison of the VSDs from the HCN1 activated and resting states, the Kv1.2/2.1 activated and resting state extracted from long timescale simulations (*Jensen et al., 2012*), and the TAX4 open structure (PDB 5H3O, *Li et al., 2017*). Small cyan arrows show the displacement of the Cα atom of R3 along the applied electric field vector. (**F**) Per-residue gating charge computed per HCN1 subunit. Positive residues are shown in blue, negative in red and non-charged in orange. Green lines show the cumulative

*Figure 1 continued on next page*

*Figure 1 continued*

gating charge (thick) and the standard error (thin). (**G**) Coupling function corresponding to the resting (cyan) and activated (red) states. The dashed lines depict the boundaries of the transmembrane part of the voltage sensor. (**H**) Hydration of the HCN1 voltage sensor in the resting (left) and activated (right) states. The shaded regions show 25-75 (dark) and 10-90 (light) percentiles of the water molecule number collected for the six voltage sensors that underwent activation.

The online version of this article includes the following figure supplement(s) for figure 1:

**Figure supplement 1.** Design and model of an HCN1 mutant with faster activation kinetics.
**Figure supplement 2.** Activation of the six voltage sensor domains observed in two independent MD simulations.

fluorometry (*Dai et al., 2019*). However, the simulations reveal an unexpected break in the S4 helix, causing the C-terminal half to adopt an orientation parallel to the membrane plane (*Figure 1A–E*, *Video 1*) when the last gating charge is transferred. Overlay of 6 independent voltage sensors confirms that they converge to the same activated (Down) structure (*Figure 1B*, *Figure 1—figure supplement 2*). This helix breaking model is in stark contrast to what has previously been observed in simulations of voltage-gated potassium channels and new resting state structures of voltage-gated sodium channels (*Delemotte et al., 2011*; *Amaral et al., 2012*; *Jensen et al., 2012*; *Vargas et al., 2012*; *She et al., 2018*; *Xu et al., 2019*; *Wisedchaisri et al., 2019*). In these channels, the consensus view is that S4 moves vertically and twists as a unit in what is typically referred to as the helical screw model (*Figure 1E*). Interestingly, the structure of the cyclic nucleotide-gated (CNG) channel TAX4 (*Li et al., 2017*), a close relative to HCN channels (*Baker et al., 2015*), shows a bent S4 helix reminiscent of what we observe for HCN1 (*Figure 1D,E*). Thus, the S4 segment in these two closely related clades may have an intrinsic propensity to bend.

The movement of S4 in response to voltage results in the appearance of gating currents, which has been measured experimentally for the sea urchin HCN (spHCN) channel but not for other HCN channels presumably due to the slow nature of their gating (*Bruening-Wright et al., 2007*; *Ryu and Yellen, 2012*). The contribution of each residue to the net gating charge can be computed from our simulations (*Figure 1F,G*) (*Roux, 2008*; *Treptow et al., 2009*). In this approach, charge contribution of each residue is calculated based on the derivative of the local electrostatic potential with respect to applied voltage in the resting and activated states (the so-called coupling function, *Figure 1G*). These contributions will only differ between states if either the charged residues move relative to the local electric field or the local electric field rearranges around these residues. Summing the contribution of each residue shows the net gating charge per VSD is 1.85 $e_o$ (or 7.4 $e_o$ per channel), which is in good agreement with the eight charges per channel suggested previously through kinetic modeling of HCN channels activation (*Hummert et al., 2018*).

As expected, the per-residue contribution to this charge difference is dominated by the two positively charged residues on S4 (R267 and K270) which move relative to the charge transfer center, with lower contributions from the other charges of S4 (*Figure 1F*). Additionally, the negative countercharges on S1-S3 contribute to the net gating charge as was experimentally shown

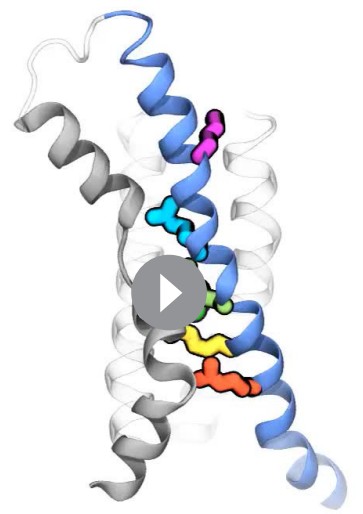

**Video 1.** Conformational changes of the HCN1 voltage sensor during activation. The S4 helix is shown in cyan, its positively charged residues in different colors (K261 purple, R267 cyan, R270 green, R273 orange and R276 red). The hydrophobic plug (F186) is colored in light green. S3 is gray and the rest of the voltage sensor is transparent.
https://elifesciences.org/articles/53400#video1

for the Shaker potassium channel (*Seoh et al., 1996*). Interestingly, the acidic residues of S1-S3 contribute to the gating charge despite little to no net displacement (*Figure 1—figure supplement 2*). This implies a redistribution of the local electric field inside the VSD upon activation, as previously suggested based on HCN1 cysteine accessibility studies (*Bell et al., 2004*). The change in the local electric field can be seen as a change in the coupling function between the activated and resting state (*Figure 1G*). The local electric field within the VSD is thought to be controlled primarily by hydration of the water accessible crevices (*Islas and Sigworth, 2001*); as expected, the hydration profile of the VSD is altered upon activation (*Figure 1H*). It is possible that the increase in hydration of the activated state comes from the mutations we introduced to accelerate gating kinetics. Alternatively, inward movement of the S4 positive charges may cause an increase in hydration of the countercharges on S1-S3 (*Seoh et al., 1996*). On the internal side, the increased hydration is likely related to the broadening of the internal crevice due to S4 bending. This increase in hydration of the VSD can be tested experimentally using substituted cysteine accessibility.

## Helical break model assessed with cysteine accessibility

In substituted cysteine accessibility, cysteines are introduced in specific locations of the channel and their solvent accessibility can be monitored using thiol reactive methanethiosulfonate MTS compounds in a state-dependent manner. From the simulations, we identified positions within the VSD which undergo a predicted change in accessibility to a sphere of 2.9 Å radius (the approximate size of MTSET (2-(trimethylammounium)ethyl methanethiosulfonate). As expected, many of these positions lie on the S4 helix and have been probed previously (*Bell et al., 2004*; *Vemana et al., 2004*). The experimentally-attained accessibility of most of these residues agrees well with the accessibility pattern predicted from our model (*Figure 2—figure supplement 1A*). A key feature of this pattern is that there is a much larger change in accessibility for the internally accessible residues than external. This indicates that there is a larger conformational change in the internal crevice and lends support to the S4 bending movement observed in our simulations. However, we sought more direct experimental evidence to rule out a canonical vertical S4 movement.

In addition to the state-dependent accessibilities in S4, the simulation also predicts an increase in the accessibility of S1-S3 upon activation. These arise due to broadening of the internally facing crevice and would not be expected if S4 moved vertically as a unit (*Figure 2A,B*, see also *Figure 2—figure supplement 1B,C*). Surprisingly, in the activated state models extracted from the simulations, even the charge-transfer center (F186) becomes accessible in 3 out of the six activated states (*Figure 2C*). As the charge-transfer center is often considered the boundary between the internal and external crevices, it is an ideal candidate to test for the crevice broadening predicted in our broken helix model. Due to the high expression levels required for assessing internal accessibility with inside-out patches, we used spHCN rather than HCN1 (*Männikkö et al., 2002*). The WT spHCN channel shows no detectable functional change following MTSET application to either the closed (depolarized) or open (hyperpolarized) states as has been shown previously (*Figure 2D* and *Männikkö et al., 2002*). In the charge-transfer center mutant (F186C), internal application of MTSET at depolarized potentials leads to very slowly evolving increase in current amplitude ($\tau_{modification}$=7.5±0.5 M$^{-1}$s$^{-1}$). This confirms that the charge-transfer center has low solvent accessibility in the resting state. In contrast, application of MTSET during hyperpolarization results in a rapid increase in peak current amplitude that quickly reaches a steady state ($\tau_{modification}$=440±30 M$^{-1}$s$^{-1}$). This more than an order of magnitude increase in accessibility of the charge-transfer center upon channel activation is difficult to reconcile with the helical screw model and supports the helix-breaking model predicted by the simulations.

## The crucial role of the S4 breakpoint serine in hyperpolarization-dependent gating

The unique voltage sensor movement we observe in MD simulations naturally raises the question about its role in determining the inverted gating phenotype of HCN channels. To explore this, we took advantage of two previously identified chimeras that only differ in the S3b-S4 voltage-sensing segment, yet show opposite gating polarities (*Figure 3A* and *Cowgill et al., 2019*). The HEHEH chimera containing the S3b-S4 voltage-sensing segment from HCN1 activates only upon hyperpolarization while the HEEEH containing the S3b-S4 voltage-sensing segment from the depolarization-

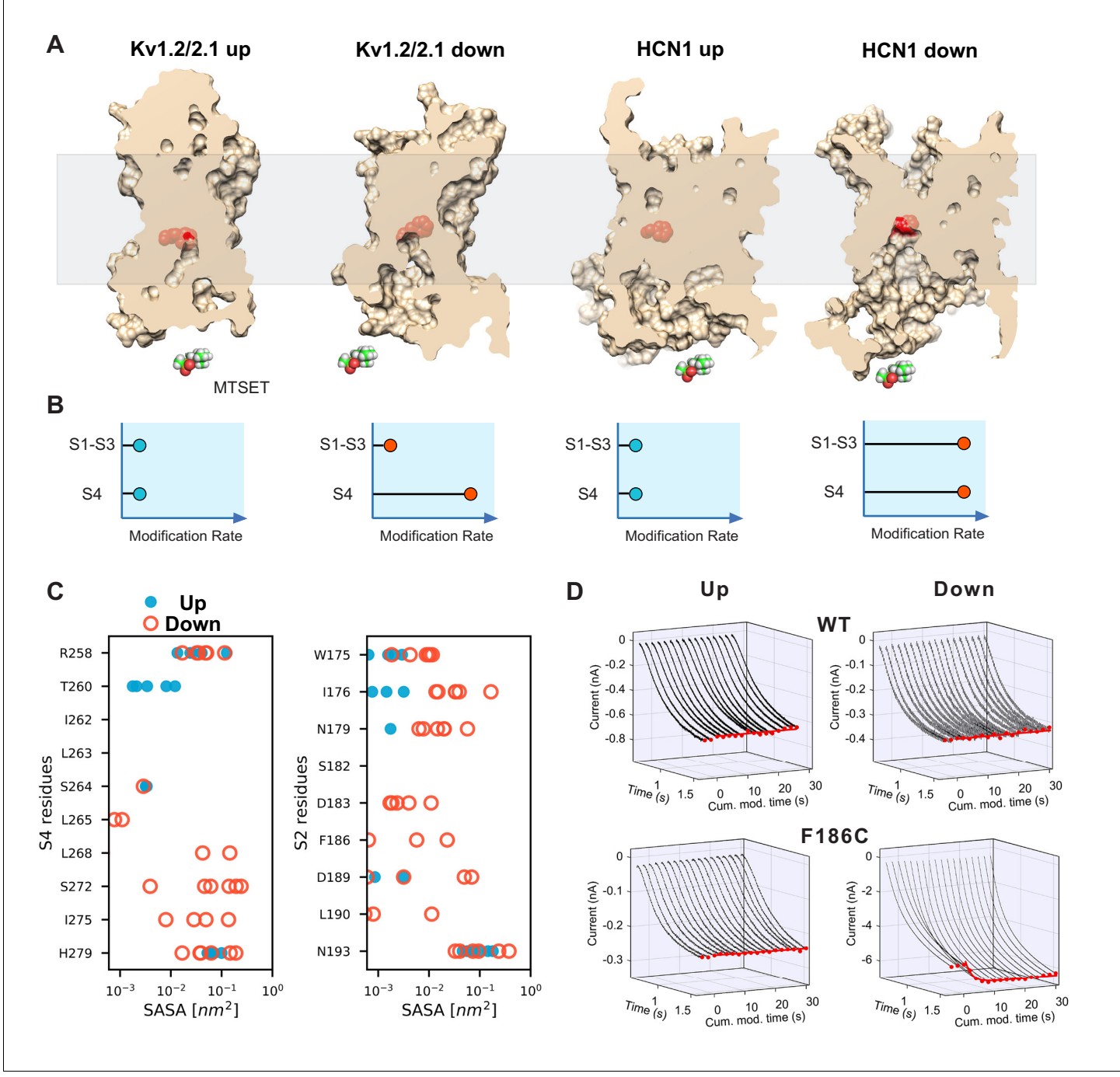

**Figure 2.** State-dependent accessibility of residues in S2 and S4 segments of HCN channel. (A) Accessibility of the S2 charge transfer center (Phe residue) in representative molecular models of the activated and resting states Kv1.2/2.1 VSDs and the resting and activated states HCN1 VSDs extracted from molecular simulations. The charge transfer center is only accessible from the intracellular medium in the activated (Down) state of HCN1. A cutaway through the VSD is represented as a slightly transparent light brown surface; the charge transfer center (Phe residue) is represented as red spheres. (B) Predicted accessibility to internally applied polar cysteine modifying reagents. Accessibility to S4 cysteines will increase in the Down state compared to the Up state for both models as has been shown previously. The S2 cysteine at or proximal to the charge transfer center is expected to show very little modification in both Up and Down states in the canonical model. However, if the helix breaks and opens a large cavity in the Down state, the cysteine at the charge transfer center will become accessible in a state-dependent fashion. (C) Solvent accessibility estimates of S4 (left) and S2 (right) residues for both up and down state models of HCN1. The accessibility of each residue is estimated for the six voltage sensors that underwent activation and shown as one symbol each (closed blue circles for the resting Up state, open orange circles for the activated Down state). (D) State-dependent accessibility of the charge transfer center of hyperpolarization-activated ion channel. Wild type spHCN channel does not react with

*Figure 2 continued on next page*

*Figure 2 continued*

MTSET under hyperpolarized or depolarized condition (top two panels). Substitution of the charge transfer center, F186C, on the other hand, shows state-dependent reactivity with MTSET (bottom two panels). The rate of reactivity in the activated Down state is $440 \pm 30$ $M^{-1}s^{-1}$ versus $7.5 \pm 0.3$ in the resting Up state.

The online version of this article includes the following figure supplement(s) for figure 2:

**Figure supplement 1.** Activated state model explains state-dependent accessibility of S4 residues.

activated EAG channel activates only on depolarization. If the helix bending movement underlies inverted gating polarity, we would predict that it is only the lower portion of S4 (lower paddle) that is required for hyperpolarization gating in these chimeras. Indeed, the chimera with only the lower paddle of HCN1 activates primarily upon hyperpolarization. At depolarized potentials, it conducts a leak current slightly larger than the HEHEH parent (*Figure 3—figure supplement 1A–C*). On the other hand, the upper paddle chimera activates only upon depolarization and its behavior closely resembles that of the parent chimera with the full EAG voltage sensor. Together, these results suggest that the source of inverted gating polarity can be localized to the lower portion of S4, precisely where we observe the helix break in our simulations.

A closer examination of the models from the simulations reveals that the S4 helix breaks in a highly consistent location centered at L271 (*Figure 3B,C*). Comparison of the consensus sequences for HCN and EAG channels show that this position is predominantly conserved between families. In fact, the overall sequence in this region is very similar between HCN and EAG with the exception of position 272. The consensus for HCN channels at this position is polar (such as serine for HCN1) whereas the consensus for EAG is hydrophobic. Furthermore, this residue is situated directly adjacent to the helix break, suggesting it may serve a vital role in channel gating. To test the role of S272 in channel gating, we introduced a series of substitutions (*Figure 3—figure supplement 1D*) in WT HCN1 channel. The lack of functional expression of mutants with hydrophobic residues at position 272 indicates that serine is critical for HCN1 channel gating but we must rule out the possibility that these channels are simply not trafficked to the membrane. As the constructs are C-terminally tagged with mCherry, we can use confocal microscopy to track surface expression. For WT HCN1, we observe mCherry surface fluorescence from oocytes displaying robust inward currents typical of HCN channels (*Figure 3—figure supplement 2A–C*). For the hydrophobic mutations, we observe surface fluorescence well above the background of uninjected oocytes, yet we still do not observe currents in these oocytes (*Figure 3—figure supplement 2A–C*). This confirms that the lack of currents detected in these mutants is due to non-functional channels and does not reflect a trafficking defect.

Functional experiments in the WT background are difficult to interpret because of confounding issues due to other structural elements in the channel. For instance, we have found that WT HCN channels rapidly inactivate upon depolarization due to the presence of tight interactions at the S4-S5 interface (*Cowgill et al., 2019*). To disentangle these effects and inspired by paddle chimera studies (*Alabi et al., 2007*), we utilized bipolar constructs such as HHHEH which can activate both upon depolarization or hyperpolarization (*Figure 3D*). We reasoned that, unlike wild type HCN channels, these channels have the gating machinery to activate in both directions and, therefore, allow us to test the relative contributions of these substitutions on outward and inward rectification. Strikingly, the S272L mutant is a predominantly outwardly rectifying channel in contrast to the parent HHHEH. We then attempted to rescue the inverted gating polarity phenotype by introducing serines throughout the surrounding region in the S272L background (*Figure 3D* and *Figure 3—figure supplement 1E*). Interestingly, only the L271S mutation was capable of restoring a primarily hyperpolarization-activated phenotype like the HHHEH parent (*Figure 3D* and *Figure 3—figure supplement 1E*). This suggests that the location of the serine in the S4 helix is critical for the inverted channel gating phenotype.

Although we were unable to observe the effect of the S272L mutation on HCN1 experimentally, we can assess its influence on channel gating in silico using free energy perturbation (FEP) (*Rodinger and Pomès, 2005*; *Zhang et al., 2019*). The free energy difference between the activation of HCN1 and its S272L mutant $\Delta\Delta G = \Delta G_{r\to a}(mut) - \Delta G_{r\to a}(HCN1)$ were calculated using FEP (*Zwanzig, 1954*) (*Figure 4*). Considering VSD-only simulations, $\Delta\Delta G$ was estimated via alchemical

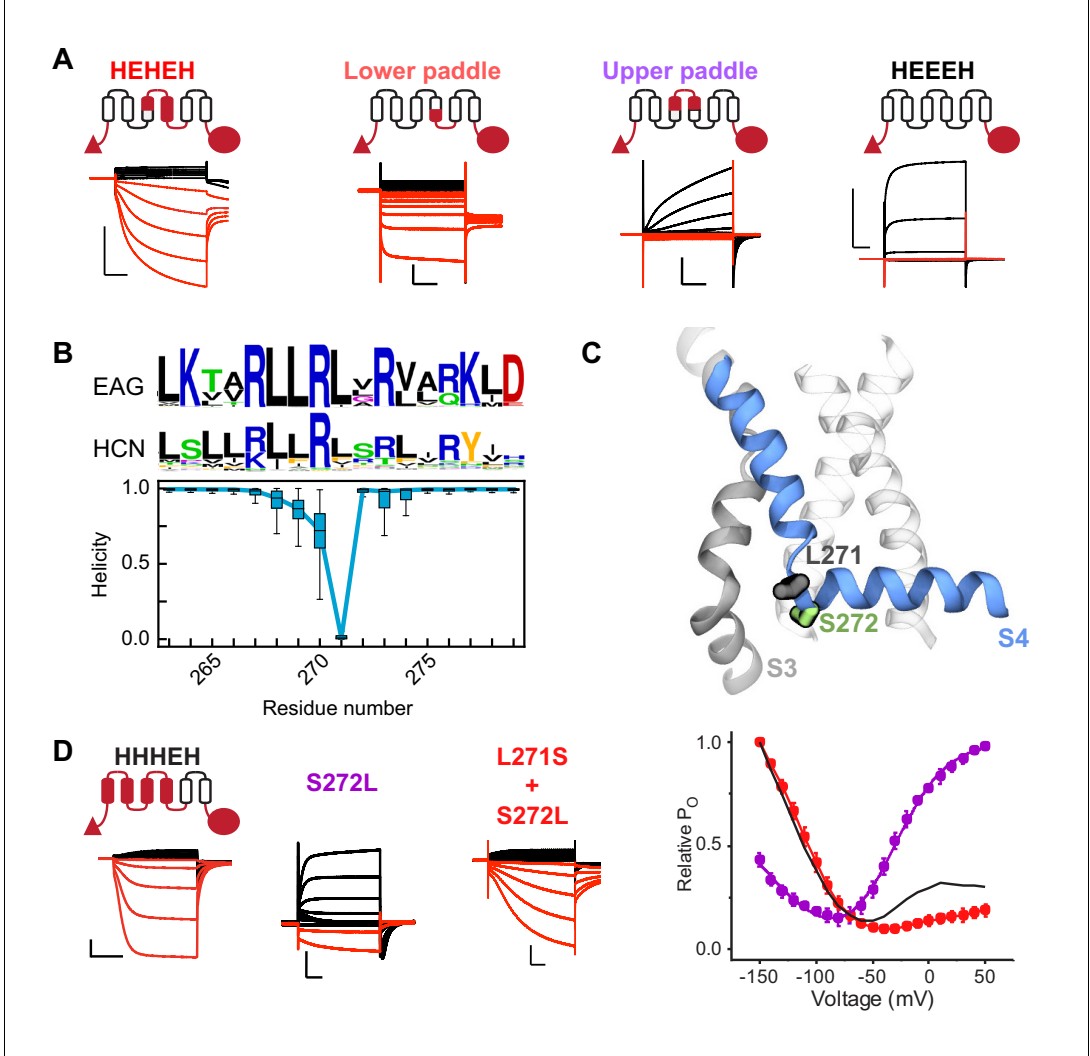

**Figure 3.** A conserved serine residue (S272) located in the lower half of the S4 segment is critical for hyperpolarization-dependent gating. (**A**) Cartoon representations and representative current traces for chimeras with varying contributions of the HCN1 S4 helix. Black traces represent current responses to depolarizing pulses whereas are red ones depict current responses elicited by hyperpolarizing potential pulses. Test pulses range from −150 mV to 50 mV from a holding potential of 0 mV (lower paddle), −50 mV (HEHEH), or −90 mV (HEEEH and upper paddle). Scale bars show 5 μA (vertical) and 200 ms (horizontal). Color coding and scale bars are same throughout the figure. (**B**) Top: Consensus sequences from multiple sequence alignments for S4 helix of EAG and HCN families shown as sequence logos (*Crooks et al., 2004*). The height of each residue is proportional to its frequency, while the height of the overall stack of residues is inversely proportional to Shannon entropy. Bottom: Helicity of S4 helix plotted as a function of residue position in the activated state of HCN1 from simulations. The box plot shows the median, 25–75% (box), 1–99% (bars) of the data collected from the six voltage sensors that underwent activation. (**C**) Structure of a representative activated state model highlighting the position of key residues near the bend (L271 and S272 in gray and green sticks, respectively). (**D**) Left: Representative current traces from the bipolar chimera HHHEH and mutants of this background near the site of the S4 bending. Test pulses range from −150 mV to 50 mV from a holding potential of −50 mV (HHHEH and L271S+S272L) or −100 mV (S272L). Right: Relative $P_O$ vs. voltage curves for the mutants. Error bars represent standard deviation n = 4 (S272L), 4 (S271S+S272L) from independent measurement.

The online version of this article includes the following figure supplement(s) for figure 3:

**Figure supplement 1.** Probing the role of HCN S4 in hyperpolarization dependent gating.
**Figure supplement 2.** Surface trafficking of HCN1 and hydrophobic mutants at S272.

paths of a thermodynamics cycle as: $\Delta\Delta G = \Delta G_a(HCN1 \rightarrow mut) - \Delta G_r(HCN1 \rightarrow mut)$. In each path S272 was reversibly transformed into a leucine through linear interpolation of the Hamiltonians of the HCN1 and mutant systems, and the free energies of these transformations were then computed using the Bennet Acceptance Ratio method (BAR) (*Bennett, 1976*). The estimated free energy

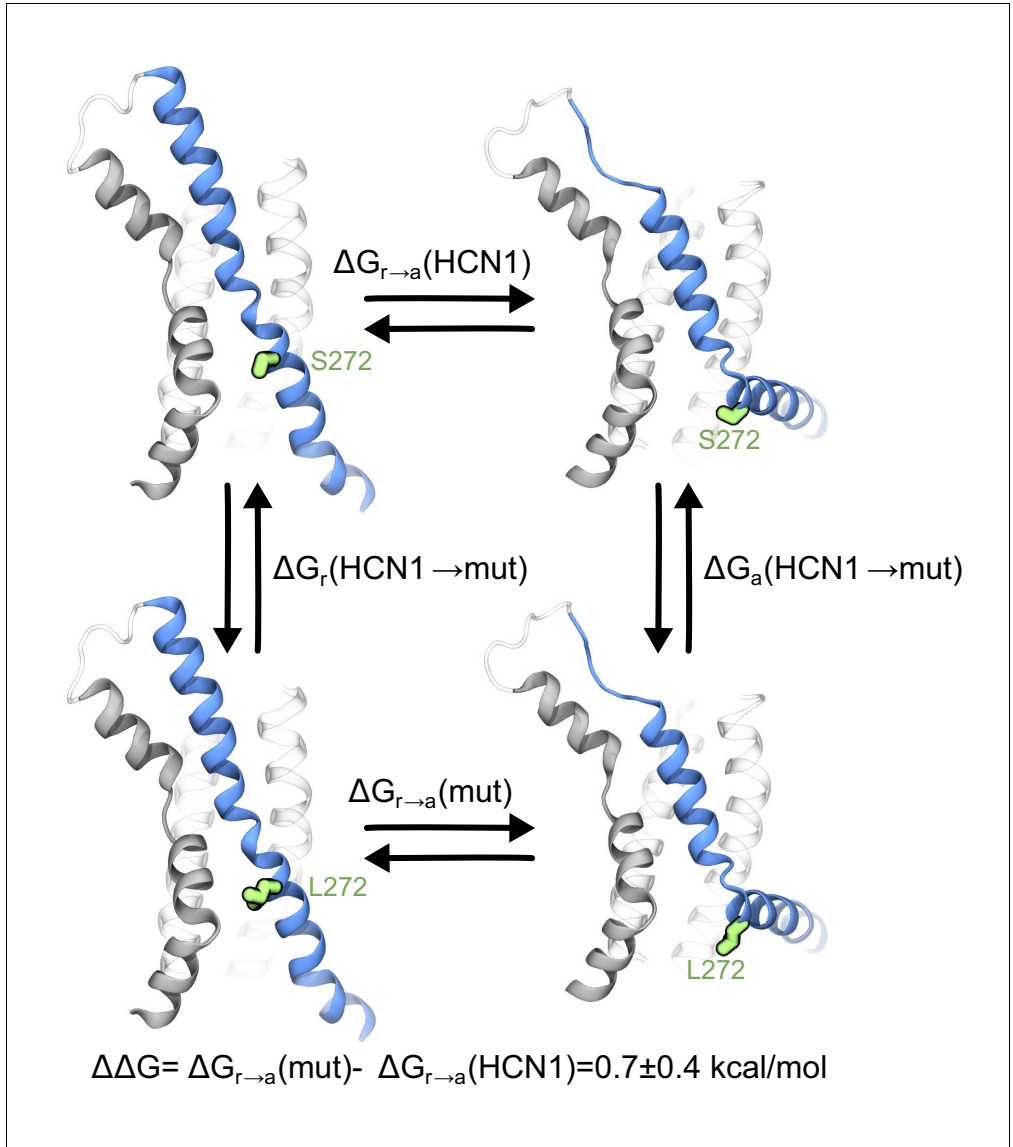

$$\Delta\Delta G= \Delta G_{r\to a}(mut)- \Delta G_{r\to a}(HCN1)=0.7\pm0.4 \text{ kcal/mol}$$

**Figure 4.** Contribution of the breakpoint serine to the free energy of activation of the HCN1 VSD. The thermodynamic cycle shows the activation of HCN1 and its S272L mutant ($\Delta G_{r\to a}(mut)$ and $\Delta G_{r\to a}(HCN1)$), and the alchemical paths of S to L transition in the two VSDs ($\Delta G_a(HCN1\to mut)$ and $\Delta G_r(HCN1\to mut)$). The mutated residue is colored in green.

difference of 0.7 ± 0.4 kcal/mol equates to ~2.8 kcal/mol per channel suggesting that the S272L mutation shifts the equilibrium toward the resting state, thus disfavoring the bending motion. The observed difference in free energy is significant because the experimentally determined coupling energy between voltage sensor and pore for spHCN is only 1.3 kT (~0.8 kcal/mol) (*Ryu and Yellen, 2012*)

## Hydrophilicity and turn propensity of the breakpoint residue determines gating polarity

We introduced a series of substitutions at S272 for different bipolar backgrounds to better understand the physicochemical underpinnings of gating polarity (*Figure 5A* and *Figure 5—figure supplement 1A–C*). Both constructs tested showed a clear trend where hydrophobic substitutions favor depolarization activation while hydrophilic substitutions favor hyperpolarization activation. To quantify the relative strengths of the hyperpolarization and depolarization activation pathways, we used

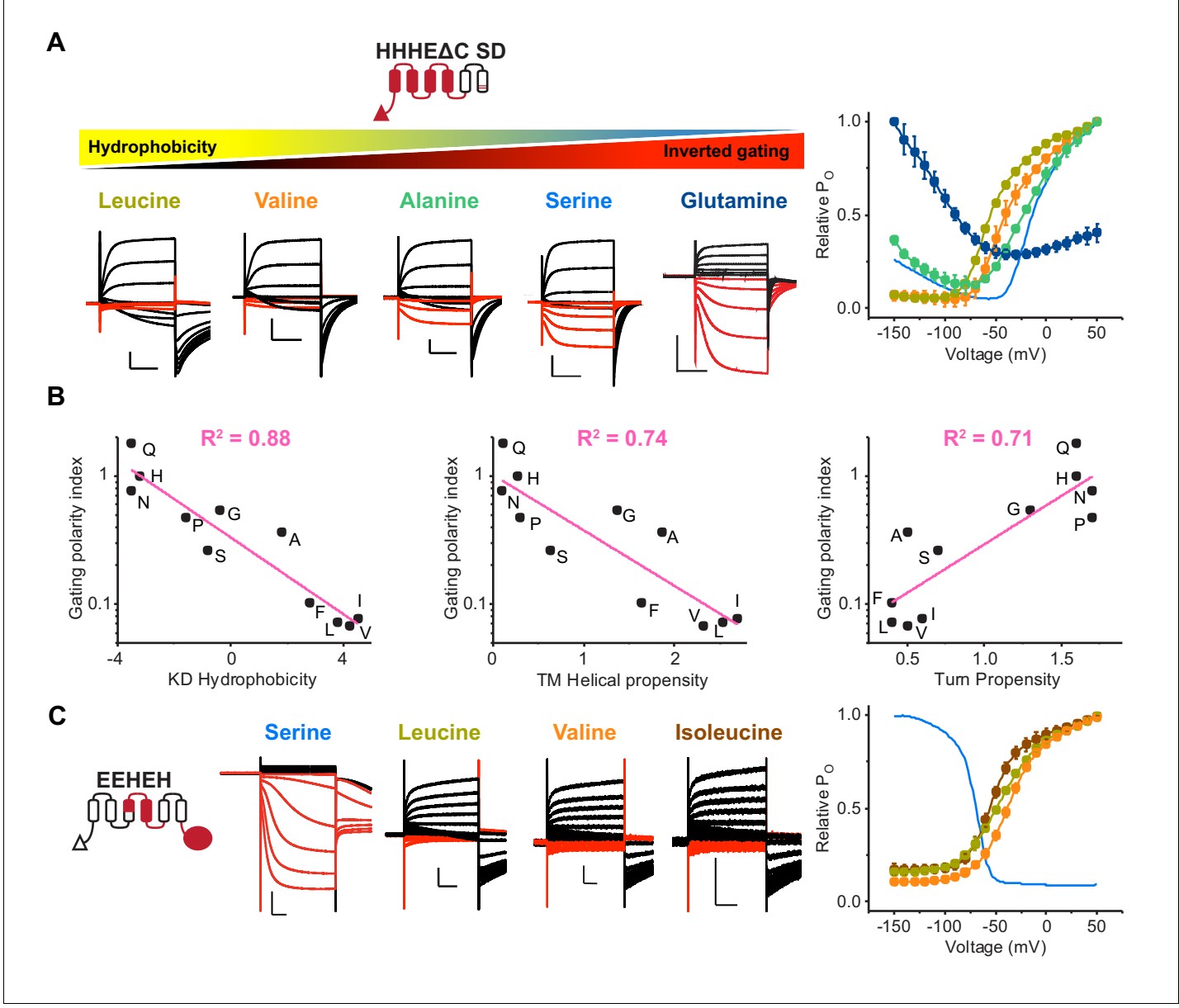

**Figure 5.** The physicochemical property of residues at the breakpoint is the primary determinant for gating polarity in HCN-EAG chimeras. (**A**) Left: Representative current traces of the parent HHHEΔC SD (serine) and S272 mutants from two-electrode voltage clamp arranged according to decreasing hydrophobicity. The SD mutations are the same described previously (*Cowgill et al., 2019*). Black traces represent current responses to depolarizing pulses whereas red ones depict current responses elicited by hyperpolarizing potential pulses. Test pulses range from −150 mV to 50 mV from a holding potential of −50 mV (Glutamine), −60 mV (Serine), or −90 mV (Leucine, Valine, and Alanine). Scale bars show 5 µA (vertical) and 200 ms (horizontal). Right: Relative $P_O$ vs. voltage curves for the mutants presented on the left. The relative Po-V curve for the parent HHHEΔC SD is shown as a blue line. Error bars represent standard deviation from n = 6 (Leucine), 5 (Alanine and Valine), 4 (Proline) from independent measurements. (**B**) Correlation of gating polarity index ($P_O^{-150\ mV}/P_O^{50\ mV}$) with Kyle-Doolittle hydrophobicity (left), helical propensity (middle), and turn propensity (right) for the HHHEΔC SD. (**C**) Left: Representative current traces of the parent EEHEH (serine) and hydrophobic substitutions at S272 position. The traces are colored as described earlier. Pulse protocol is from −150 mV to 50 mV from holding potential of −40 mV (EEHEH) or −100 mV (all others). Scale bars show 2 µA (vertical) and 200 ms (horizontal). Right: Relative $P_O$ vs. voltage curves for the mutants presented on the left. The relative Po-V curve for the parent EEHEH is shown as a blue line. Error bars represent standard deviation n = 5 (Leucine and Isoleucine), 4 (Valine) from independent measurement.

The online version of this article includes the following figure supplement(s) for figure 5:

**Figure supplement 1.** Bipolar chimeras point to the outstanding role of S272 for hyperpolarization dependent gating.

the ratio of the open probability at negative compared to positive membrane potentials ($P_O^{-150\ mV}$/$P_O^{50\ mV}$). This parameter, which we refer to as the gating polarity index, is inversely correlated with residue hydrophobicity with a tight fit (*Figure 5B*). Previous work on model helices showed that the helical propensity is directly correlated to hydrophobicity in membrane environments (*Li and Deber, 1994*). Therefore, the negative trend we observed between gating polarity index and hydrophobicity may be caused by the helix-stabilizing nature of hydrophobic residues in transmembrane segments. Indeed, the strong, negative correlation is upheld when comparing the gating polarity index to an experimentally-derived scale for transmembrane helical propensity (*Liu and Deber, 1998*). Finally, the gating polarity index shows positive correlation to the independently-derived turn propensity (*Monné et al., 1999*), which has previously been used as a metric for helix breaking proclivity. Together, these correlations offer strong support for our helix breaking model and reveal that residue polarity at S272 is central to the gating mechanism of HCN channels.

Given the strong effect of the S272 mutants on gating phenotype, we wondered whether introduction of a serine at the equivalent position in EAG channels would be sufficient to confer hyperpolarization activation. The G362S mutant of hEAG1 shows depolarization-dependent opening with no signs of hyperpolarization activation (*Figure 5—figure supplement 1D*). This is not entirely surprising as both HCN and EAG channels have presumably evolved multiple structural elements to favor opening on either hyperpolarization or depolarization. Therefore, we hypothesized that a chimera with minimal components derived from HCN1 (EEHEH) would be especially sensitive to mutations of S272. We refer to this channel as a 'proto-HCN' chimera because it shares some properties of both HCN and EAG channels, which have evolved from a common ancestor. Remarkably, the leucine, valine, and isoleucine mutations in this background eliminate all detectable hyperpolarization activation in the EEHEH background, transforming an HCN-like parent into an EAG-like channel with a single point mutation (*Figure 5C*). Thus, divergence of hyperpolarization-activated channels from depolarization-activated channels could have occurred through a single point mutation in the S4 segment.

## Discussion

The activation mechanism of HCN channels (*Bell et al., 2004*; *Männikkö et al., 2002*; *Vemana et al., 2004*) involves a movement of S4 analogous to that observed in depolarization-activated channels (*Larsson et al., 1996*; *Ahern and Horn, 2005*), but the mechanisms which ultimately lead to inward rectification remain poorly understood (*James and Zagotta, 2018*). Here, we have resolved the activation pathway of the HCN1 voltage-sensing domain at the atomistic level using multi-microsecond long molecular dynamics simulations starting with the full-length HCN1 resting state structure. We observe that in response to a hyperpolarizing electric field, the S4 positive charges slide down by approximately two helical turns. Coupled to this motion is a ~ 80° bend in S4, which leads the bottom half of this helix to orient itself almost parallel to the membrane plane. Our down state model of the S4 voltage-sensor is consistent with existing S4 cysteine accessibility (*Bell et al., 2004*; *Vemana et al., 2004*) and transition metal FRET data (*Dai et al., 2019*). However, the predicted state-dependent accessibility of the residues in the gating scaffold is incompatible with the classical helical screw motion observed in voltage-gated potassium and sodium channels (*Figure 2—figure supplement 1C*), see also *Pathak et al. (2007)*. Our studies show that the charge transfer center of HCN channels exhibit a distinct pattern of accessibility consistent with the molecular simulations. Furthermore, we find that this helix breaking transition is necessary for inward rectification and our studies show that the gating polarity strongly correlates with the energetics of transmembrane helix formation. Strikingly, in a proto-HCN background, the nature of the residue in the S4 breakpoint is the central determinant of gating polarity and the direction of rectification can be switched by a single amino acid substitution.

The mechanism of voltage-sensor activation has been a focus of many studies (*Larsson et al., 1996*, *Chanda et al., 2005*; *Posson et al., 2005*; *Long et al., 2005*; *Swartz, 2008*; *Delemotte et al., 2011*; *Amaral et al., 2012*; *Henrion et al., 2012*; *Jensen et al., 2012*; *Kintzer et al., 2018*; *Li et al., 2015*; *Bezanilla, 2018*; *Wisedchaisri et al., 2019*) over the past two decades, and the emerging consensus in the field is that VSD activation involves a helical screw motion across the membrane relative to the surrounding S1-S3 bundle which acts as a gating scaffold. In all these channels, the measured gating charge originates from the physical displacement of

S4 in a relatively static electrostatic environment which is also conserved across different voltage sensor domains (*Souza et al., 2014*). During the gating process, step-wise rearrangement of salt bridges between S4 basic residues and acidic countercharges on S1-S3 (*Tao et al., 2010*; *Pless et al., 2011*) facilitates charge transfer from one side to the other. The extent of the S4 motion and the number of rearrangements (termed 'clicks') it undergoes depends, however, on the channel and the magnitude of the electrical force exerted on the S4 positive residues. In part due to lack of structural information, most of the mechanistic studies were focused on outward rectifying ion channels and very little was known about the mechanism of inward rectification. Our simulations based on the high-resolution structure paints an entirely different picture and seems to indicate that at least another type of activation mechanism exists, in which the downward motion of S4 is accompanied by a break in the center of the helix and a change in the electrostatic environment. Whether this activation mechanism is a hallmark found only in HCN channels or is also present in related families remains to be elucidated. Of particular interest now is to understand the activation mechanism in non-domain swapped ion channels activated by depolarization such as hERG and EAG (*Trudeau et al., 1995*) and examine the role of cytosolic domains in modulating voltage-dependence of activation (*Xia et al., 2002*; *Miranda et al., 2018*)

Until recently, all voltage-gated ion channels were also thought to be domain-swapped, with the voltage-sensing domain of one subunit sitting adjacent to the pore domain of its neighboring subunit. Allosteric coupling between the VSD and the pore domain were thought to involve direct contacts between the S4-S5 linker with the C-terminal end of S6 located at the intracellular gating interface (*Lu et al., 2002*; *Soler-Llavina et al., 2006*; *Long et al., 2005*; *Muroi et al., 2010*; *Haddad and Blunck, 2011*). The other gating interface involves contacts between the transmembrane residues in the S4 and those in the neighboring S5 segment (*Hou et al., 2017*; *Fernández-Mariño et al., 2018*; *Carvalho-de-Souza and Bezanilla, 2019*). Recently, several Kv channels including HCN1, EAG and hERG were shown to adopt a non-swapped architecture wherein the VSD of a given subunit is placed next to the pore domain of the same subunit, in an assembly that obviates the need for a S4-S5 linker helix (*Lee and MacKinnon, 2017*; *Wang and MacKinnon, 2017*; *Whicher and MacKinnon, 2016*). Our results provide clues towards elucidating the coupling mechanism between the VSD and the pore domain in HCN channels. While we did not observe pore opening in our simulations, we noticed a consistent motion of S5 in all channel subunits (*Figure 6A,B*). The S5 N-terminus was displaced away from its original position (*Figure 6A*), and towards one closer to the putative open conformation of TAX4 and hERG channels (*Figure 6B*). Such hinge-like conformational switches are often observed in simulations of channel pores and are thought to be a common mechanism of channel gating (*Forrest and Sansom, 2000*). An analysis of contacts maintained throughout the simulations showed that the VSD and S5 communicate via two interaction networks (*Figure 6C*). When the lower S4 sub-helix torques towards the membrane, the S4/S5 interactions at the lower gating interface causes the lower part of the S5 to tilt while the top of the S5 remains anchored in place due to interactions with the relatively static S1 segment at the upper gating interface (*Figure 6D*). The conservation of contacts explains how spHCN functions as a split channel in which the covalent link between S4 and S5 is severed (*Flynn and Zagotta, 2018*). We speculate that this splaying of S5 from the central pore in an iris-like motion provides S6 with the space needed for pore opening.

### Note added in proof

While this manuscript was in press, the structure of the HCN channel with voltage-sensor trapped in the down state (*Lee and MacKinnon, 2019*) appeared online. The down state structure shows that the S4 helix breaks and the bottom half of the S4 forms an interfacial helix. Their structure is consistent with our findings described here and provides compelling evidence that the HCN S4 adopts a unique conformation in the down state.

## Materials and methods

### System preparation for long-timescale molecular dynamics simulation

The structure of the HCN1 apo state (pdb code 5U6O) was used as a starting structure for both Anton runs. Missing residues were built using Modeller (*Sali and Blundell, 1993*). Additionally, the

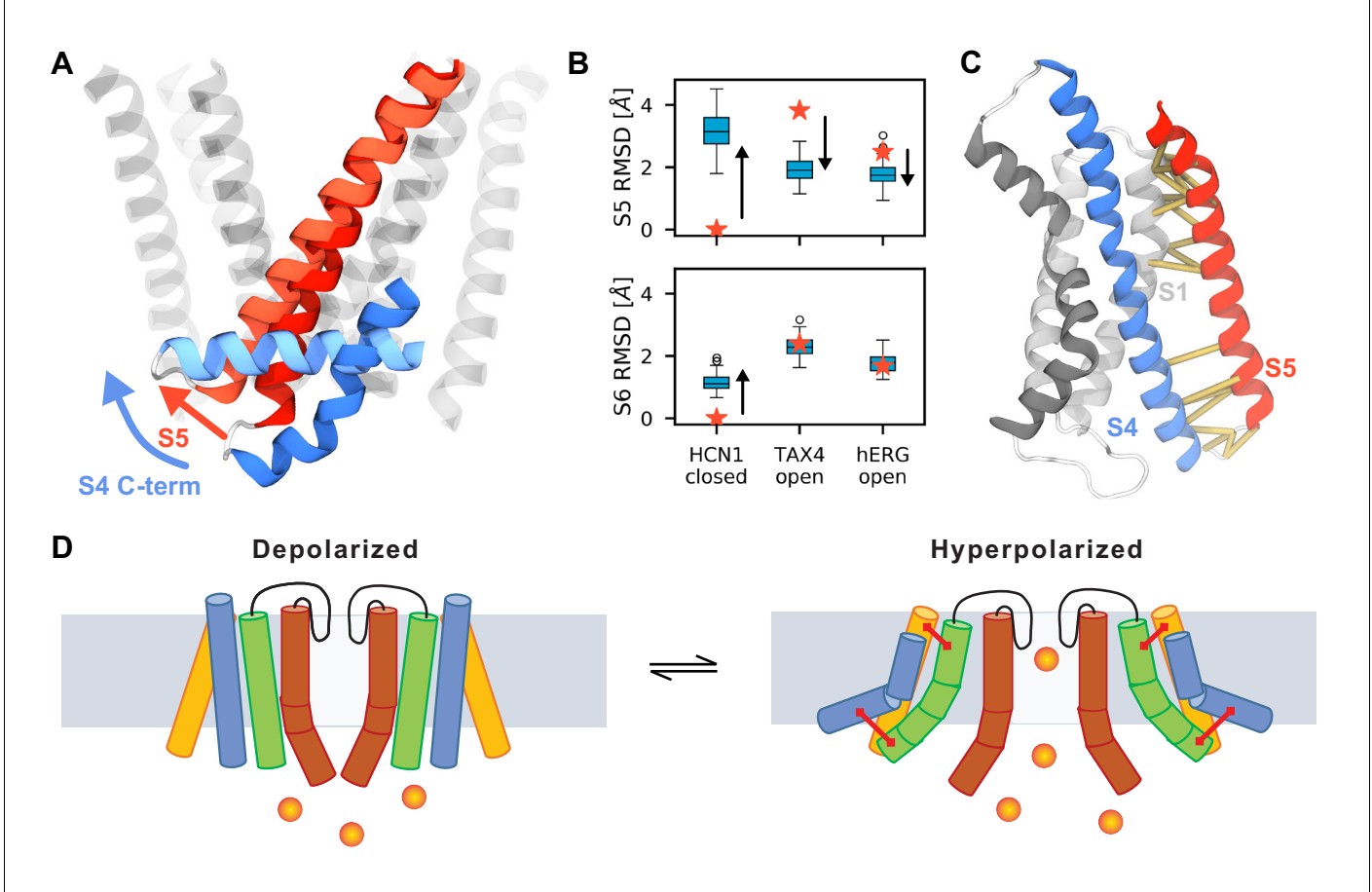

**Figure 6.** Possible mechanism of hyperpolarization-dependent opening. (**A**) Conformational rearrangements of S4 and S5 during activation. At the end of the simulations, the C-terminus of S4 (shown in dark blue for the resting Up state and in light blue for the activated Down state) is parallel to the membrane which causes the S5 helix (dark red for the resting Up state and in light red for the activated Down state) to tilt at the bottom. S5 helices of other subunits and S6 are shown as transparent helices for context. (**B**) Root mean square deviation (RMSD) of the S5 (top) and S6 (bottom) of HCN1 before (stars) and after (boxplots) MD simulations. HCN1 closed shows the RMSD values with respect to the initial structure of HCN1 (PDB 5U6O); TAX4 open shows the RMSD values with respect to the structure of TAX4 (PDB 5H3O); hERG open shows the structure of RMSD values with respect to the structure of hERG (PDB 5VA1). The arrows indicate the direction along which the HCN1 conformation evolves during activation. For instance, in case of S5, it diverges from the conformation in the HCN1 structure, and approaches that in TAX4 and hERG. (**C**) Network of interactions between the voltage sensor and the pore domain observed in the MD simulations (yellow), separate into two groups: between the C-termini of S1 and S5, and between the C-terminus of S4 and the N-terminus of S5. (**D**) Suggested model for coupling between the voltage sensor and the pore domain of HCN1. Upon activation, S4 (blue) pulls the N-terminus of S5 (green) through the first interaction network. The other interaction network anchors the C-terminus of S5 to a static S1 (yellow). These motions create room for S6 rearrangement and allow it to relax to an open conformation.

structure was refined before the second run due to instabilities of several protein domains observed during the first run (see the structure refinement section below). The system for molecular dynamics simulations was prepared using the CHARMM-GUI server (*Jo et al., 2008*). Briefly, the structure of the HCN1 apo state was embedded in a 1-palmytoyl-2-oleoyl-phosphatidylcholine (POPC) bilayer and initially solvated in a 150 mM KCl solution. The ions were later removed during the first Anton run, and before the second Anton run, after we noticed that they promoted unpacking of the voltage sensor domains. The CHARMM36 force field (*Mackerell et al., 2004*) was used for the protein, lipids and ions, and the TIP3P model for water. The M153T/I160V mutant was built using the mutator plugin of VMD (*Humphrey et al., 1996*).

## Molecular dynamics simulations

The initial steps of the WT and M153T/I160V mutant equilibration were performed on the PDC supercomputer Beskow using Gromacs 2018.1 (*Abraham et al., 2015*). During these steps (~100 ns

in total) the restraints applied to the protein were gradually released first from the sidechains and then from the backbone. The simulations were performed in the NPT ensemble; Nose-Hoover thermostat (*Nosé, 1984*) and Parrinello-Rahman barostat (*Parrinello and Rahman, 1981*) were used to keep the temperature and the pressure constant at 300 K and 1 bar; the timestep was set to 2 fs.

The ~3 μs runs of the WT and the M153T/I160V mutant systems under an electric field were performed on Beskow using Gromacs 2018.1 (*Abraham et al., 2015*). The transmembrane potential was set to 750 mV using the external electric field method. Other simulations parameters were kept identical.

The M153T/I160V mutant system was then transferred to Anton (*Shaw et al., 2014*) and further equilibrated for ~1 μs. Additional restraints were applied to the hydrogen bonds between the backbone groups of the S4 C-terminus prior to the second long run. These restraints were required to stabilize the secondary structure of the S4 C-terminus that was otherwise partially lost in one of the subunits during the first run. They were completely released after the 600 ns simulation. The simulations were performed in conditions of constant temperature (300 K, Nose-Hoover thermostat [*Nosé, 1984*]) and pressure (1 bar, MTK barostat [*Zhang et al., 2017*]). The multigrator approach was used to coherently update the thermostat every 24$^{th}$ step, barostat every 480$^{th}$ step and Newtonian particle motion (*Lippert et al., 2013*). The timestep was kept to 2 fs.

After the equilibration, we gradually increased the transmembrane potential to −750 mV, and to −550 mV in the first and second runs, respectively. In the first run, we later decreased it to −550 mV after ~1.5 μs. The overall length of the trajectory under an electric field was ~18.8 and 22.8 μs for the first and second runs, respectively. The simulations under transmembrane potential were performed in the NVT ensemble. Other parameters were duplicated from the equilibration.

## Structure refinement

During the first Anton run we observed that the HCN domain, the selectivity filter and the A' helix of the C-linker deviated from their initial positions significantly. This instability may result from a relatively low local resolution structure (of the HCN domain specifically), absence of stabilizing factors (cAMP or phosphatidylinositol 4,5-bisphosphate PIP$_2$ lipids, for instance), dynamic properties of these domains, or other factors. While we cannot rule out all of these possibilities, we were able to test whether alternative modeling of the aforementioned domains in the regions with ambiguous electronic density improved their stability in molecular dynamics simulations during the second Anton run. To do so, we refined the structure using Phenix real-space refinement (*Adams et al., 2010*) and the cryo-EM map (EMD-8512); 4-fold symmetry was imposed using non-crystallographic symmetry restraints. Overall five macro-cycles were performed with global minimization, local rotamer fitting and simulated annealing. The resulting model was then locally refined using Coot (*Emsley and Cowtan, 2004*). Briefly, several conformations for the domains regions where the electronic density is ambiguous were generated and refined using real space refinement; torsion, planar peptide and trans peptide restraints were applied, and the refinement weight was set to 40. The model with the best density fit and the smallest number of Ramachandran outliers was further used for the final refinement in Phenix. Simulated annealing was not considered and the remaining settings were identical to that in the first Phenix refinement. The resulting structure was used as a starting point for the second Anton run and is deposited as a part of Supplementary Material.

During the second Anton run the HCN domain and the A' helix of the C-linker were still unstable indicating that their instability potentially finds its origin elsewhere rather than in the modeling of the 3-d structure into the cryo-EM density. These domains partially lost their secondary structure shortly after the restraints on the backbone have been released, similar to what happened in the first Anton run. The selectivity filter, on the other hand, remained stable for a much longer time in the second Anton run (~10 μs compared to only few ns in the first Anton run). After ~10 μs, however, Y361 rotated toward the channel axis and blocked the conductive pore. Similar collapse of the selectivity filter has been observed in other potassium channels at low ionic concentration or in the absence of ionic current (*Cuello et al., 2010*; *Zhou et al., 2001*) and has previously been proposed as a mechanism of voltage-dependent gating in K2P channels (*Schewe et al., 2016*). We hypothesize that while the additional refinement did increase the stability of the selectivity filter other factors such as low ionic concentration or the absence of ionic current resulted in its collapse.

## Construction of the multiple sequence alignment

The initial hidden Markov models of the HCN and EAG transmembrane regions were built based on HCN and EAG sequences extracted from the Swiss-Prot database. They were then submitted to HMMER (*Eddy, 1995*) and iteratively updated with newly identified homologous sequences. Overall 176 and 266 sequences homologous to HCN and EAG channels were found. The resulting multiple sequence alignments were finally used to build sequence logos (*Crooks et al., 2004*).

## Calculation of the gating charge

The per-residue and total gating charge was computed using the 'coupling function' method (*Roux, 2008*; *Treptow et al., 2009*). In this method, the total gating charge is decomposed into individual contributions from residues as $Q = \sum_j Q^j$, and each contribution is further expressed in terms of the so-called coupling function $f$:

$$Q^j = \sum_i q_i^j [f_a(r_i^j) - f_r(r_i^j)], \tag{1}$$

$f_a(r_i^j)$ and $f_r(r_i^j)$ in *Equation (1)* represent dimensionless coupling of the charge $q_i^j$ to the transmembrane potential $V_m$ in the activated (*a*) and resting (*r*) states, and the summation runs over all charges $q_i^j$ of the $j^{th}$ residue. The coupling function is then approximated as the rate of change of the local electrostatic potential $\varphi(\boldsymbol{r})$ with respect to $V_m$:

$$f_a = \frac{\partial \varphi(\boldsymbol{r})}{\partial V_m} \approx \frac{(\varphi(\boldsymbol{r}, V_{m1}) - \varphi(\boldsymbol{r}, V_{m2}))}{V_{m1} - V_{m2}}, \tag{2}$$

where $V_{m1}$ and $V_{m2}$ are two different transmembrane potentials.

In practice, from each Anton run under an electric field we extracted ten conformations: five for the resting state and five for the activated one. For every conformation nine two ns molecular dynamics simulations under different transmembrane potentials were performed. The local electrostatic potential was then computed for each obtained trajectory using the PME Electrostatics plugin of VMD (*Humphrey et al., 1996*). The resulting values were further combined to obtain the coupling function for each extracted conformation. Finally, the coupling functions were used to compute the per-residue and total gating charge through *Equation (1)*.

## Calculation of per-residue helicity

Per-residue helicity $hp$ was estimated based on the $\varphi$ and $\psi$ torsions:

$$hp = (1 + \cos(\phi - \phi_\alpha))(1 + \cos(\psi - \psi_\alpha))/4, \tag{3}$$

where $\varphi_\alpha$ and $\psi_\alpha$ are corresponding torsions in a perfect α-helix.

## Calculation of solvent accessible surface area (SASA)

Solvent accessible surface area (SASA) was calculated using Gromacs 2018.1 SASA (*Abraham et al., 2015*). Radius of the solvent probe was set to 2.9 Å according to the size of MTSET. Renderings presented in all Figures except 2 and S5 were prepared using Visual Molecular Dynamics VMD (*Humphrey et al., 1996*).

## Free energy calculations

Free energy perturbation (*Zwanzig, 1954*) was performed on the voltage sensor domain only (residues 140–290) on Beskow using Gromacs 2018.1 (*Abraham et al., 2015*). Representative conformations of the resting and activated states were extracted from the trajectories under an electric field, embedded into the system with a POPC bilayer and water, and equilibrated for 100 ns. For the resting state, 3 conformations were extracted from the equilibration trajectories while for the activated state we considered 5 conformations spanning its structural diversity. For every conformation the S272→L transformation was performed through linear interpolation of the Hamiltonians of the systems with HCN1 voltage sensor and its mutant; λ was used as a coupling parameter such as when λ = 0 the Hamiltonian corresponded to the HCN1 system, while when λ = 1 it was of the S272L system. Overall, 21 consecutive λ windows were considered; in each, the system was first equilibrated

for 4 ns and then the production run was performed during 8-20 ns. The free energy difference and the corresponding error were calculated using the Bennet acceptance ratio method (BAR) (*Bennett, 1976*) (*Supplementary file 1*). The free energy difference between the activation of HCN1 and its S272L mutant and the corresponding error are calculated as the average and the standard error based on the 3 to 5 independent estimations.

## Molecular biology

The chimeric constructs were generated between mHCN1 and hEAG1 in pUNIV vector as described previously (*Cowgill et al., 2019*). Note that although the mouse isoform is used here, we maintain the human HCN1 numbering throughout the figures and text for consistency with the structure and simulations. Additionally, the charge transfer center mutation (F186C) refers to hHCN1 numbering for clarity, which is equivalent to position F260 in spHCN. Site directed mutation was done by Quick Change mutagenesis reaction using a standard protocol. The coding sequences of all constructs were verified using Sanger sequencing. Constructs were linearized with SbfI overnight (15 hr) and transcribed using the T7 mMessage Kit according to the protocol. For experiments requiring inside-out patches, the T7 Ultra kit was used for transcription. RNA was isolated by lithium chloride precipitation and washed with 70% ethanol according to the manufacturer's protocol then resuspended to 1–2 mg/ml using RNase free water.

## Recombinant expression and electrophysiology

Oocytes were surgically removed from adult *Xenopus laevis* under anesthesia in accordance with the protocol approved by the Animal Care and Use Committee of Wisconsin- Madison at University of Wisconsin-Madison. Stage V-VI oocytes were prepared by treating with 1 mg/ml collagenase A (Roche) in a calcium- free ND96 solution for 45 mins to 1 hr until the removal of follicular membrane. The oocytes were stored in ND96 solution with calcium containing BSA (ND96 solution: 96 mM NaCl; 2.5 mM KCl; 1 mM MgCl2; 5 mM HEPES; 1.8 mM CaCl2; pH 7.40). Oocytes were maintained in ND96 solution before injection and then transferred to ND96 containing antibiotics (50 µg/ml gentamicin and ciprofloxacin, 100 µg/ml tetracycline, penicillin, and streptomycin) and BSA (0.5 mg/ml) after injections called the post injection solution. Oocytes were injected with 20–90 ng of mRNA using Nanoject II (Drummond Scientific). Injections were done in ND 96 without calcium chloride solution. After injection oocytes were kept in the post-injection solution as mentioned above. Two-electrode voltage-clamp (TEVC) recordings were obtained at room temperature with an OC-725C amplifier (Warner) at a sampling rate of 10 kHz. Thin-walled glass pipettes (World Precision Instruments) were used with tip resistances of 0.3 to 0.8 MΩ filled with 3 M KCl. The external solution contains 100 mM KOH, 5 mM NaOH, 20 mM HEPES and 2 mM EGTA. Solutions were adjusted to pH 7.4 using methanesulfonic acid. All recordings were obtained with no leak subtraction. Data in Relative $P_O$ vs. voltage curves were fitted to a sum of two Boltzmann curves in Origin 2017 (OriginLab) with the function $I(V) = O_1 + (A_1 - O_2)/(1 + \exp(k_1(V - V_1))) + O_2 + (A_2 - O_2)/(1 + \exp(k_2(V - V_2)))$, where $A_1$ and $A_2$ represent the amplitudes, $O_1$ and $O_2$ represent the offsets, $V_1$ and $V_2$ represent $V_{1/2}$, and $k_1$ and $k_2$ represent the slope factors for two independent components. As many curves do not reach saturation, curves are primarily provided for visual reference.

## Cysteine accessibility experiments

Ionic currents were measured from inside-out patches excised from *Xenopus* oocytes after 1–3 days of mRNA injection. Currents were recorded with an Axopatch 1D amplifier, low-pass filtered at 2 kHz and sampled at 10 kHz. Pipettes were made from borosilicate glass with resistance ranging between 0.7–1.0 MΩ when filled with pipette solution (in mM): 120 KCl, 10 HEPES, and 1.0 CaCl₂, pH 7.2. The intracellular solution contained (in mM): 55 KCl, 5 KF, 10 $K_4O_7P_2$, 0.1 $Na_3VO_4$, 10 $K_2EGTA$, 10 HEPES, diC₈ PIP₂ 100 nM, pH 7.2. KF, 10 $K_4O_7P_2$ and 0.1 $Na_3VO_4$ (phosphatase-inhibitors) and diC₈ PI(4,5)P₂ were used. The 1 mM MTSET was prepared by diluting from a 1 M stock prepared daily and stored on ice in internal solution prior to the experiment. Inside-out patches containing sp-HCN channels were exposed to MTSET via a rapid perfusion system in which a computer-controlled valve was used to switch between the internal solution without or with MTSET 1 mM. The dead time of the perfusion apparatus was determined as previously described (*Oelstrom et al., 2014*) and it was typically 150 ms. This dead time was considered to make the

MTSET perfusion coincide with the pulses that open and closed the channels. The time constant of MTSET modification was calculated by fitting peak current at the end of the hyperpolarizing pulse versus the cumulative exposure time to a mono-exponential function. The accessibility (calculated as the apparent second-order rate constants) of MTSET to each cysteine mutant was determined by dividing the reciprocal of the product of the time constant by the MTSET concentration.

## Surface expression experiments

Surface fluorescence was measured on a Leica SP8 3X STED Super-resolution microscope using 576 nm excitation and collecting emission from 590 to 650 nm with a 10X, 1.4 NA objective. Detector gain, pinhole diameter, Z-position, integration time, and all other instrument settings were kept constant between all oocytes measured. Surface fluorescence was defined as the integrated intensity within a box of fixed dimension using ImageJ. Following confocal measurements, oocytes were maintained individually to enable correlation of current recordings to confocal measurements. Currents were recorded in two electrode voltage clamp as described above with the exception that ND96 solution was used externally.

## Quantitative and statistical analysis

Clampfit (Molecular Devices) was used to quantitate currents at steady state. Origin was used to fit data points to a sum of two Boltzmann curves. Kinetic Model Builder was used for all kinetic simulations. PyMOL was used for the structural analysis in *Figure 5—figure supplement 1D* and USCF chimera for the one in *Figure 2*. Throughout the paper, n is used to denote the number of oocytes tested in each experiment as indicated in each figure legend.

## Acknowledgements

This work was supported by grants from the Gustafsson Foundation and Science for Life Laboratory to LD; National Institutes of Health to BC (NS101723), JC (T32 HL-07936–18), JL (T32 GM008293); Anton 2 allocation to BC (PSC17025P and PSC18025P); Romnes faculty fellowship to BC; Science and Medicine Graduate Research Scholars (SciMed GRS) fellowship to JL. Preliminary simulations and free energy calculations were performed on resources provided by the Swedish National Infrastructure for Computing (SNIC) at the PDC Centre for High Performance Computing (PDC-HPC). Anton 2 computer time was provided by the Pittsburgh Supercomputing Center (PSC) through Grant R01GM116961 from the National Institutes of Health. The Anton 2 machine at PSC was generously made available by DE Shaw Research. The authors also thank N Nallappan and A Thangaraju for help generating chimeras and A Thangaraju, N Nallappan, and W Stevens-Sostre for performing frog surgeries. We thank Y Itoh for help with the structure refinement and RJ Howard for assistance in preparing *Figure 2*. We would like to thank GA Robertson for providing hEAG1, C Czajkowski for providing the pUNIV vector, B Santoro and SA Siegelbaum for providing mHCN1, and UB Kaupp for providing spHCN. We would like to thank S Chowdhury and other members of the labs for helpful discussions and input throughout the project.

## Additional information

### Competing interests

Baron Chanda: Reviewing editor, *eLife*. The other authors declare that no competing interests exist.

### Funding

| Funder | Grant reference number | Author |
|--------|------------------------|--------|
| National Institute of Neurological Disorders and Stroke | NS101723 | Baron Chanda |
| National Heart, Lung, and Blood Institute | HL-07936-18 | John B Cowgill |
| National Institute of General Medical Sciences | GM008293 | Jenna L Lin |

The funders had no role in study design, data collection and interpretation, or the decision to submit the work for publication.

## Author contributions
Marina A Kasimova, Conceptualization, Data curation, Formal analysis, Validation, Visualization, Methodology; Debanjan Tewari, Data curation, Formal analysis, Validation, Investigation, Methodology; John B Cowgill, Conceptualization, Data curation, Formal analysis, Investigation, Visualization, Methodology; Willy Carrasquel Ursuleaz, Data curation, Formal analysis, Investigation, Methodology; Jenna L Lin, Data curation, Formal analysis, Investigation; Lucie Delemotte, Conceptualization, Supervision, Funding acquisition, Validation, Project administration; Baron Chanda, Conceptualization, Supervision, Funding acquisition, Project administration

## Author ORCIDs
Marina A Kasimova (iD) https://orcid.org/0000-0002-7497-9448
John B Cowgill (iD) https://orcid.org/0000-0002-7968-8359
Lucie Delemotte (iD) https://orcid.org/0000-0002-0828-3899
Baron Chanda (iD) https://orcid.org/0000-0003-4954-7034

## Decision letter and Author response
Decision letter https://doi.org/10.7554/eLife.53400.sa1
Author response https://doi.org/10.7554/eLife.53400.sa2

# Additional files

## Supplementary files
• Supplementary file 1. Free energy differences for different replicates of the S to L alchemical transformations of the HCN1 VSD.

• Transparent reporting form

## Data availability
Simulations were carried out at the Pittsburg Supercomputing center which is funded by NIGMS. Data is available at: https://psc.edu/anton-project-summaries?id=3071&pid=30.

The following dataset was generated:

| Author(s) | Year | Dataset title | Dataset URL | Database and Identifier |
|---|---|---|---|---|
| Marina A Kasimova, Debanjan Tewari, John B Cowgill, Willy Carrasquel Ursuleaz, Jenna L Lin, Lucie Delemotte, Baron Chanda | 2019 | Non-canonical voltage-sensor pore coupling in the hyperpolarized cyclic nucleotide gated channel | https://psc.edu/anton-project-summaries?id=3071&pid=30 | Pittsburgh Supercomputing Center, 3071 |

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
