## [Decision Letter]

**Acceptance summary:**

Some voltage-gated ion channels are opened by hyperpolarizing (more negative) membrane potentials and others open with depolarization. This differential gating polarity of related voltage sensitive channels has been the subject of intense investigation both because of its physiological implications and because understanding the structural and biophysical mechanisms determining the different behaviors will reveal basic principles of voltage gating. This paper reports a combination of functional studies and molecular computations that proposes and supports a novel structural mechanism of voltage sensing and gating.

---

## [Author Response]

[Editors' note: we include below the reviews that the authors received from another journal, along with the authors’ responses.]

Reviewer #1:

Major points:1) Authors should consider eliminating the term "paddle" throughout the manuscript. It is a specific term meaning helix-turn-helix entity with structural and functional implications. In the context of this paper the function or structure is not shown.

We appreciate the feedback but in this context the use of the “paddle” is appropriate. It is widely used to define the S3b-S4 helix and has been shown to be a portable module. In fact, this terminology has been used to describe the primary voltage-sensing region in many channels even when high-resolution structures of those channels were not available (Bosmans et. al. (2008) Nature 456, 202-208).

2) This reviewer has major concern regarding the length of the simulations. Longest simulations of ~350 µs have been done for Kv1.2/2.1 paddle chimera from David Shaw's Lab (Jensen et al., Science, 2012). In essence, the authors can improve the quality and length of simulations as per this reference. Reason being, MD simulations of 20 µs is too short for HCN. For HCN channels fastest family member requires milliseconds to open as measured in cells. I am concerned that simulations might not reflect the correct opening based on time constraints.

As mentioned in the manuscript we have used two strategies to speed up the activation of HCN1, which, as pointed out by the reviewer, is known to open on a millisecond timescale.

The first strategy was to apply a large electric field (-550 mV), similar to what has been done by Jensen et al. for Kv1.2/2.1. The ~350 µs simulations which the reviewer refers to were conducted to observe both deactivation and activation of the channel and were performed under a lower electric field (in an adaptive manner, varying between -375 to -500 mV, making it difficult to infer the average value used; see Jensen’s paper Figure 2A, B and Supplementary file 1).

Importantly, under a larger electric field (varying between -375 to -750 mV), the full deactivation of Kv1.2/2.1 (all four voltage sensors) was observed within ~ 65 µs (see Jensen’s paper Figure 1D). In our simulations, we used the voltage of -550 mV throughout the simulation #2 and starting from 1.5 µs in the simulation #1. We anticipate that the application of the large electric field greatly accelerated the activation of HCN1 bringing it from milliseconds to tens of microseconds timescale. In particular, extrapolation of the HCN1 opening time constant to strongly hyperpolarized voltages indeed suggests that it can be as low as ~ 20 µs (see Author response image 1). Using this approach, the time constant for Shaker return from the activated state to the resting state is predicted to be from 18.5 µs (at -500 mV) to 134 µs (at -375 mV), which is consistent with the timescale of observed movement by Jensen et al.,2012.

**Author response image 1. respfig1:** Kinetics of HCN1 and Shaker activation. Left: The logarithm of the activation time constant (in milliseconds) is plotted against voltage for HCN1 and fit to a line. The time constant for activation at -550 mV is calculated using the equation shown. Right: The logarithm of the time constant (in ms) for off gating charge movement for Shaker is plotted against voltage. The predicted time constants at -375 and -500 mV are calculated as in the left panel.

The second strategy we used was to perform the simulations of an HCN1 mutant with increased hydration of the voltage sensors, as suggested by Lacroix and Bezanilla, 2013, Neuron. These authors indeed showed that substitution of a hydrophobic residue by a hydrophilic one in voltage-gated potassium channels can speed up their activation up to 3 times. We therefore anticipate that this strategy further allowed us to shift the HCN1 activation time constant towards the timescales available to MD simulations performed on Anton. As controls, we have shown that the activation of the HCN1 mutant follows a similar path than that of the WT (Figure 1—figure supplement 1B of the original manuscript; in both channels the activation starts with the displacement of S4 charges along an electric field), and that the designed mutant is fully functional in electrophysiology experiments, which argues against the possibility that it has an altered activation mechanism. Finally, we have shown that our simulations are in excellent agreement with the already published experimental data (such as puzzling solvent accessibility of S4, the FRET experiments, etc.) and, importantly, provide novel predictions that were further tested in our study.

Finally, we would like to point out that only DE Shaw research can carry out hundreds of microseconds long simulations because no one has access to their special purpose machine outside of the one computer that was donated to PSC. This limits simulations to tens of microseconds at most for any individual research project.

3) Collapse of the filter is observed in the simulations. However, the electrical currents shown did not inactivate or diminished throughout the recordings. If the filter was collapsing the measured function of HCN should be altered (as it happened in KcsA and the authors mentioned the reference Cuello et al., 2010). There is discrepancy between simulations and electrophysiology experiments. How reliable are simulations to guide the experiments when the filter collapse in 10 µs and the currents did not inactivate?

The simulations here were performed for the closed structure of HCN1 (which remained closed throughout the MD runs), and therefore we cannot anticipate whether we would see the same collapse of the selectivity filter in the open state under conditions of ionic current. Based on this, we do not follow the reviewer’s argument and see no discrepancy between the simulations and the experimental data mentioned by the reviewer.

Additionally, the reviewer is probably aware that it is common practice to restrain the selectivity filter in molecular simulations to avoid collapse. The collapse is generally attributed to the inability of classical force fields to describe polarizability. This phenomenon is well known to be extremely specific to the issue of conduction of ions through the selectivity filter of potassium-selective and related cation-selective channels and has no bearing on all the other aspects of channel function, including voltage sensor domain activation.

4) Furthermore, authors state "Similar collapse of the selectivity filter has been observed in other potassium channels at low ionic concentration or in the absence of ionic current (Cuello et al., 2010; Zhou et al., 2001) and has previously been proposed as a mechanism of voltage-dependent gating in K2P channels (Schewe et al., 2016)."To the best of my knowledge in this study the KCl was 150 mM as mentioned in the Materials and methods. Then why we see the filter collapsed if low potassium was not tested?

There appears to be some misunderstanding here. We actually use low ionic strength condition during our simulations and it not clear to us why the reviewer believes that we used 150 mM KCl. Therefore, it is not surprising that the selectivity filter also tends to collapse in our simulations. The relevant section from the text is copied below:

"Briefly, the structure of the HCN1 apo state was embedded in a 1-palmytoyl-2-oleoyl-phosphatidylcholine (POPC) bilayer and initially solvated in a 150 mM KCl solution. The ions were later removed during the first Anton run, and before the second Anton run, after we noticed that they promoted unpacking of the voltage sensor domains. "

5) Figure 3—figure supplement 1D: This reviewer is concerned that the lack of currents for Leucine, Isoleucine, and Valine could be due to altered surface expression, as also mentioned by the authors in the explanation. There is a concussive way to show whether constructs express at the plasma membrane and if yes then indeed the mutants are non-functional.

We provide experimental data to show that the mutants are trafficked to the surface membrane. See Figure 3—figure supplement 2A and B.

Minor points:1) Was the net gating charge of HCN measured previously or only suggested? In its current form it is not clear. If suggested then on what basis? "The reference net gating charge per VSD is 1.85 e (or 7.4 e per channel), which is in good agreement with the 8 charges per channel suggested previously (Hummert et al., 2018)."

The net gating charge of HCN2 channel was calculated previously by fitting electrophysiological behavior to a detailed kinetic model. The original language was unclear and we have clarified the sentence to reflect how the experimental charge determination was made.

2) There should be some discussion/explanation that why 6 out of 8 simulations were successful. "Substantial conformational change was observed in 6 out of 8 voltage sensors from two independent ~20 µs simulations of the full-length channel (Figure 1A, B)." Is it norm of this kind of experiments or technical challenge?

Molecular dynamics simulations predict the evolution of a system of particles along time. As such, they reproduce the stochasticity of motions that biomolecular system naturally undergo. The events observed in these simulations occur thus with a certain probability which increases with the length of simulations. With a longer MD run, we would thus be able observe the activation of all 8 voltage sensor domains. The stochasticity of MD simulations is also well documented in other computational works, such as the one by Jensen et al. mentioned in this reviewer's point 2. For instance, out of five simulations performed on the resting/closed Kv1.2/2.1 channel under depolarizing electric field, only two showed substantial conformational changes in the voltage sensor domains (see Jensen’s paper Supplementary file 1 simulations 10-14). Moreover, in those two simulations, the gating charges responded only in one (two) VSD for the first (and the second, respectively) simulation.

Reviewer #2:

Is it possible that the hydrophilic 'breakpoint' residue is required not to break the S4 helix but to allow the lipid head group/glycerol backbone to move to that position? In that case, a hydrophobic amino acid substitution would interact with the acyl-chain, incorrectly positioning the lipid molecule and altering gating. The accessibility change would then be explained by movements of the lipids and not from breaking of the helix.

We are not sure that we understand the reviewer’s concern because it is not clear how the proposed mechanism can explain a complete switch in the gating polarity from inward rectifying to outward rectifying. Interaction with a single phospholipid may account for subtle changes in gating properties but it is difficult to imagine how it can lead to the observed phenotype.

Additionally, we are unaware of any experimental evidence to suggest that lipid interactions at S272 are involved in gating. Although HCN channels have been shown to be regulated by PIP_2_ levels in the membrane, this regulation has been primarily shown to involve interactions with the C-linker within the carboxy terminus of the channel.

Most importantly, if a lipid headgroup is involved, then the interactions would be electrostatic in nature and would depend on the charge of the residue. However, we observe no difference between polar residues and charged residues. Moreover, proline which lacks any charge but promotes helix disruption promotes inverted gating phenotype like the serine at that position. We should also point out that we observed strong correlation between turn propensity and gating polarity. To the best of our knowledge, such correlations have never been observed when considering interactions between lipid molecule and amino acid side-chains.

The cysteine accessibility studies are used to highlight that F186C is exposed in the HCN1 down state, which is inconsistent with helical screw model. It would be helpful to have a series of positions that show that the helical-breaking model is correct instead of primarily showing that the helical screw model is not correct. For example, the incompatibility with the helical screw model may arise from the misplacement of lipids in the proposed accessibility model (shown in the figure) instead of where they are actually located in the real membrane.

We thank the reviewer for this comment, we agree that conclusions should generally not be reliant on the solvent accessibility of a single position. However, solvent accessibility experiments on the S4 helix of HCN1 had been already performed by two independent research groups in 2004. In Figure 2—figure supplement 1, we compare the predicted accessibility from our simulations to 11 positions tested in these studies. Our model correctly predicts the accessibility pattern of 9 of the 11 residues tested in those studies. It is worth noting that for the one of the two residues that are not consistent, we do not observe accessibility in the simulations for a position with very low experimentally determined accessibility. This may indicate that we were unable to sample timescales long enough to observe accessibility. More importantly, both experimental accessibility studies have previously shown that there are larger state-dependent accessibility changes for the internally facing S4 residues than for externally facing residues. This is consistent with our model showing that the S4 helix breaks in the down state and broadens the inside facing crevice. .

We agree that it is worth considering alternative explanations for our experimental results. However, we would like to point out that it is the steric occlusion within the voltage sensing domain that limits accessibility of F186 as well as the S4 helix. We have attempted to highlight this in in the surface representations of Figure 2A and in cartoon form in Figure 2—figure supplement 1. This occlusion is observed in the experimentally determined resting state. Therefore, the drastic changes in accessibility we (and others) show experimentally must be caused by substantial conformational change of the protein itself. Without knowing more details, it is hard to envision how rearrangement of the lipid around the protein could explain the changes in accessibility observed given the structure of the resting state of HCN1.

From a philosophical standpoint, we agree with the reviewer that we cannot rule out a more complex model involving additional parameters such as lipid molecule etc. Given the available data, our model involving helix breaking transition is the most parsimonious. Our experiments are designed to distinguish between existing models. As the helical screw model is the canonical model of voltage sensor movement, we elected to focus on distinguishing it from other proposed mechanisms. The only other model that has been proposed which would be consistent with our observed accessibility change to the charge transfer center is the so-called ‘transporter model’ in which S4 is relatively static while S1-S3 rearrange around it. Indeed, Seigelbaum and colleagues in their accessibility study favored this model over the canonical sliding helix type of model. Although the transporter model could be consistent with the accessibility observed by us and others, it would fail to explain why hyperpolarization activation is sensitive to mutations at S272. This position faces towards the pore and away from S1-S3 in the resting Up state. Given the strong agreement of our model with the accessibility patterns shown by us and others, as well as the structure-function data at the break point, we feel that we have strong experimental evidence supporting the S4 breaking model.

The sequence conservation suggests that other HCN channels will exhibit the same helix-breaking in S4. Are similar structural changes observed with these channels?

A similar solvent accessibility pattern of the S4 residues has been reported for spHCN, suggesting that this channel employs the same helix-breaking mechanism. In addition, in the original manuscript, we refer to the example of a CNG family member (TAX4, see Figure 1E) which is homologous to HCNs and whose structure was resolved in the open state. Remarkably, in this channel, the S4 helix is broken in the same position as we find in HCN1.

The FEP calculations of the serine (272) to leucine (272) transition support their model. Similar calculations upon mutating neighboring positions, such as those shown in Figure 3—figure supplement 1, should be made to further test their model and the appropriateness of the FEP calculations in this context.

Indeed, performing such additional simulations may give more confidence in the appropriateness of the FEP calculations. However, these calculations are computionally expensive and to do this throughly for various sites is beyond the scope of this manuscript.

Mutations are used to increase the rate of activation of the channel for the Anton simulations (M153T/L157V). What are the properties of the M153T/L157V mutant using electrophysiology?

We thank the reviewer for this comment. Both the simulations and the electrophysiology experiments (Figure 1—figure supplement 1) were performed for M153T/I160V. We apologize for the numbering error in the original manuscript and have changed the manuscript accordingly.

It's unclear whether Figure 1—figure supplement 1 (M153T/I160V) is an electrophysiology experiment or a computational calculation. This mutation isn't discussed in the main text, but M153T/L157V is. These mutations are near (in 3-dimensional space) to residue 272 (the break helix). Could these mutations be responsible for the observed helix break?

We thank the reviewer for this comment. Please see the response to the previous comment for the first part of the question. For the second part, we do not think that M153T and I160V are responsible for the S4 break. Though we were not able to run a control simulation of appropriate length without these mutations (see response to reviewer 1 for a discussion of the rationale for using such mutants), our experiments supporting the idea of the S4 break were performed on HCN1 and chimeras lacking these mutations. Again, there are also previously reported solvent accessibility and FRET experiments performed on WT spHCN channels that are compatible with our model.

In the text, the relationship between depolarization/hyperpolarization and inward/outward rectification needs to be stated for the general reader.

We agree this connection was not properly explained in the text and have clarified in revisions.

Jargon such as "minimal HCN background" should be explained in the text.

We now refer to these as “proto-HCN channels”. Because the HCN and EAG channels are related, we think of these as some type of ancestral channels which have some properties of both HCN and EAG channels.